# Domiciliary Carers’ Perspectives on Alcohol Use by Older Adults in Their Care: A Systematic Review and Thematic Synthesis of Qualitative Studies

**DOI:** 10.3390/ijerph21101324

**Published:** 2024-10-06

**Authors:** Catherine Haighton, Mel Steer, Beth Nichol

**Affiliations:** 1Department of Social Work, Education and Community Wellbeing, Northumbria University, Newcastle upon Tyne NE7 7XA, UK; bethany.nichol@northumbria.ac.uk; 2Department of Nursing, Midwifery and Health, Northumbria University, Newcastle upon Tyne NE7 7XA, UK; m.steer@northumbria.ac.uk

**Keywords:** domiciliary carers, alcohol, older adults, care, systematic review, thematic synthesis, qualitative

## Abstract

As global populations continue to age, alcohol consumption rises, and we strive to age in place, it is important to have an up-to-date understanding of domiciliary carers’ perspectives on older adults’ alcohol use in their care. Therefore, a systematic review and thematic synthesis of qualitative studies of the unique challenges faced by domiciliary care workers in front line roles regarding older adults’ alcohol use was conducted (PROSPERO registration number: CRD42024516660). Eight databases were searched on 22 February 2024 for qualitative studies focusing on older adults’ (defined as aged 50 or over) alcohol consumption and domiciliary care. The Critical Appraisal Skills Programme checklist was utilised for quality appraisal. Twenty articles reporting 14 unique studies of mainly medium to low quality were included. Three overarching themes (and associated subthemes) were identified as follows: identification (alcohol problems are common, no assessment for alcohol problems, and additional overt signs of excessive alcohol use), management (to buy or not to buy that is the question, balancing rights and risks, monitor and report but do not intervene, maintaining the vicious circle, home as a barrier to accessing support and services, and more support needed from healthcare professionals), and training (lack of alcohol education). Domiciliary carers are well placed to make every contact count to target alcohol consumption but would benefit from support and resources for alcohol consumption identification and management. Clear guidance on how to manage alcohol consumption to harmoniously balance rights and risks is crucial, particularly when caring for older adults with cognitive difficulties.

## 1. Introduction

Global population ageing is taking place and is expected to continue over the coming years [1]. It is important to understand how this demographic shift will impact public health. A particular area of interest involves the use of alcohol among older adults (defined as those aged 50 and over (https://www.ageuk.org.uk/ (accessed on 29 September 2024)). Alcohol consumption ranges from abstinence to heavy use and it is difficult to establish where, along this continuum, problematic use begins. Excessive alcohol consumption is associated with serious social, psychological, physical, and economic costs [2]. Alcohol is a risk factor for cancers, coronary heart disease, high blood pressure, liver cirrhosis, pancreatitis, and stroke [3]. As people age, physiological changes mean that older adults are more sensitive to the effects of alcohol, and experience problems at lower levels of consumption [4]. Alcohol and medication interactions are common for this age group [5,6], with the average person aged 50 or above taking at least four prescribed medications a day [7] and with alcohol a major contraindication for many of these drugs. Combined alcohol and medication use is estimated to affect up to 19% of older Americans [8], while drinking alcohol for medicinal purposes is also prevalent [6]. Excessive alcohol consumption has also been associated with impairments in the Instrumental Activities of Daily Living (IADL) [9] and can contribute to the onset of dementia and other age-related cognitive deficits [10], Parkinson’s disease, and a range of psychological problems including depression and anxiety [11]. Alcohol use is implicated in one-third of all suicides in the older population [12].

However, moderate alcohol consumption has been associated with a lower risk of dementia [13] and functional decline [14], a reduced risk of stroke [15], protection against heart disease [16], and lower overall mortality [17]. Additionally, alcohol use among older adults provides clear social benefits including the facilitation of social interaction and leisure activities [18]. Alcohol consumption in later life is not static either, and reasons for changes in consumption vary by age, sex, and socio-economic position [19]. Research has also shown that older adults use alcohol for a number of different reasons [20]. For example, stressful life events, such as bereavement or retirement, may trigger late-onset drinking in some [21,22]. Alcohol use in older adults has also been associated with self-medication for both physical and mental health problems [6] as well as insomnia, and has also been linked to boredom, loneliness, isolation, and homelessness [21]. However, the direction of causality in the relationship between alcohol use and many of these factors is often in doubt [20], and clearly the relationship between alcohol use and health is complex. In recent years there has been a small but steady increase in the amount of alcohol consumed by older age groups [23].

The total number of people aged over 65 years requiring support with at least one task of everyday living, such as eating or dressing, is estimated at 2.2 million [24]. Twenty-four percent of men and 28% of women aged 65 and over need help with at least one Activity of Daily Living (ADL) (such as getting up and going to bed, using the toilet, and bathing) over a month, and 21% of men and 29% of women need help with at least one IADL (such as paying bills, shopping, and doing housework or the laundry) [24]. The proportions needing help with ADLs or IADLs increases with age from 21% of adults aged between 65 and 69 to 52% of those aged 80 and over [24]. Improvements in standards of living and health mean people are often able to live independently much later in life; however, people’s needs become increasingly complex in old age as more and more people live with multiple long-term conditions [25,26]. Despite a decreasing ability to live independently, older adults mostly display a preference for the ability to age in place rather than in residential settings, increasing the need for both formal and informal care [27].

This emphasis on care in the community means that domiciliary carers (including both formal paid carers and informal carers) are well placed to respond to problems with alcohol use among older people, but research suggests that they experience both structural and personal barriers to adopting a more active role [28]. Herring and Thom proposed that the provision of appropriate information and training may go some way towards encouraging carers to remain alert to alcohol-related problems in older clients; however, this hypothesis was proposed almost three decades ago [28]. A more recent review of health and social care providers’ perspectives on older people’s drinking [29] identified few studies, published up to mid-2018, which focussed on domiciliary carers. Bareham et al. reported that domiciliary care providers recognised the social opportunities associated with drinking amongst older clients, accepted their drinking, and sometimes supported clients in purchasing alcohol [29]. However, given the broad inclusion criteria for care settings, a synthesis of the unique challenges faced by domiciliary carers was minimal and relatively little of the research evidence gathered views from care workers in front line roles [29]. Thus, in the context of global population ageing and continued reliance on care in the community, the aim of this study was to gain an up-to-date understanding of domiciliary carers’ perspectives on older adults’ alcohol use in their care.

## 2. Method

### 2.1. Design

This systematic review was registered with the PROSPERO international prospective register of systematic reviews (https://www.crd.york.ac.uk/prospero/display_record.php?ID=CRD42024516660 (accessed on 22 February 2024)) and is reported according to the Preferred Reporting Items for Systematic Reviews and Meta-Analyses (PRISMA). Prior to pre-registration, PROSPERO, Joanna Briggs Institute Registries, and the Open Science Framework were searched for reviews of a similar scope, of which none were identified.

### 2.2. Eligibility Criteria

The SPIDER framework (Sample, Phenomenon of Interest, Design, Evaluation, Research type) for qualitative reviews was used to inform eligibility criteria. Domiciliary carers, both formal (e.g., paid carers) and informal (e.g., family and friends), of older adults aged 50 and over of any gender were included. Studies where the age range of older adults fell below 50 were included if the mean age was 50 years of age or above. Studies focusing on a range of drinking histories (i.e., from occasional drinkers to dependence) were included. Professionals such as general practitioners, nurses, physiotherapists, occupational therapists and social workers who typically only provide one-off, short-term care in the home or home visits were excluded. Included studies needed to examine domiciliary carers’ experience of and attitude to alcohol consumption by older adults in their care. Studies that also included their experience of and attitude to other substances such as tobacco were only included if the data were separable by substance. If studies included both home and residential care settings, the studies were included if data were identifiable by setting. Studies were included if they were qualitative in design. Where mixed methods were employed, consideration was given to inclusion if qualitative data were relevant and sufficient to the review question (the quantitative data were not included). Quantitative studies were excluded. The principal outcome of interest was key themes and direct quotations relating to current attitudinal evidence regarding how domiciliary carers view alcohol use in older adults in their care. Studies published in the peer-reviewed or grey literature were included, although books or other reviews, conference abstracts, editorials, opinion pieces, and commentaries were excluded. Backwards citation searching was applied. Whilst studies published worldwide were considered, only those published in the English language were included as resources were not available for translation. There was no restriction on the year of publication.

### 2.3. Search Strategy

Medline, ASSIA, APA PsycArticles, the Nursing and Allied Health Database, the Psychology Database, the Public Health Database, ProQuest Dissertations and Theses Global via ProQuest, and CINAHL via EBSCO were searched on 22 February 2024. Keywords relating to older adults, home care, alcohol use, and qualitative studies, each separated by the Boolean operator “AND” were searched in the title and abstract. Searches were limited to the English language and humans. Search terms including truncations, wildcards, and limits are outlined in Appendix A.

### 2.4. Study Selection

All references were uploaded to Rayyan and duplicates were removed. Two reviewers (CH/BN) independently screened titles and abstracts against the eligibility criteria followed by a screening of the full text of potentially relevant studies. Reasons for exclusion during full-text screening were recorded. The screening process has been made publicly accessible via Rayyan (https://rayyan.ai/reviews/987359 (accessed on 29 September 2024) and https://rayyan.ai/reviews/941122 (accessed on 29 September 2024)). Interrater reliability was calculated at each stage by calculating Cohen’s Kappa [30] and applying the conservative parameters set by Altman [31]. Disagreements were discussed and resolved by a third reviewer (MS).

### 2.5. Quality Assessment

The quality of each included study was examined independently by one reviewer (CH), with a 10% sample examined by a second reviewer (BN) using the Critical Appraisal Skills Programme (CASP) tool for qualitative research. The quality was reported but was not used as a basis for exclusion. To facilitate the meaningful incorporation of study quality, acknowledged as an important component of qualitative evidence synthesis, quality judgement was weighted according to CASP items five (”was the data collected in a way that addressed the research issue?”) and eight (”was the data analysis sufficiently rigorous?”) which were considered to be particularly relevant to the review’s aims. Specifically, in accordance with Cochrane guidance [32] and in the absence of evidence-based standardised criteria [33], each study was categorised into ”low” (neither item is addressed), ”medium” (one item is addressed), and ”high” (both items are addressed) quality.

### 2.6. Data Extraction

A standardised data extraction form was used independently by one reviewer (CH), with a 10% sample extracted by a second reviewer (BN) to collect the following data: the study reference, the location and context of the study, methodology, participant details, headline findings, the number of quotes, and author conclusions. In addition, all direct quotations were extracted from each paper where available. A bespoke data extraction form was developed in accordance with the specific aims of the current review, using guidance from Cochrane [32]. In accordance with open science practices, the completed data extraction Excel file was uploaded onto the Northumbria University Research Repository (https://figshare.northumbria.ac.uk/articles/dataset/Domiciliary_carers_perspectives_on_alcohol_use_by_older_adults_in_their_care_Data_extraction_for_a_systematic_review_and_thematic_synthesis_of_qualitative_studies/26064721/1 (accessed on 20 June 2024)).

### 2.7. Data Synthesis

Free coding was applied line by line to both direct quotations and interpretations by the authors of the included studies. Codes were descriptive initially, then were developed into analytical themes. Codes reflected second- and third-order constructs described in meta-ethnography; codes that were described by authors of included studies, and codes identified by the review team to describe patterns and differences across included studies, respectively [34]. In accordance with a reflexive approach to thematic analysis [35], independent coding was not conducted [36]. Instead, all analysis was conducted by the lead author (CH) and discussed with the research team to reflect upon and refine themes and sub themes.

## 3. Results

Initially, 6221 articles were identified and, after the removal of duplicates, 4279 were screened for eligibility (see Figure 1 for a PRISMA diagram). After the exclusion of 4244 articles based on the title and abstract, 35 articles were screened for eligibility based on the full text, of which 14 articles were included. Substantial agreement was achieved for inter-rater reliability, with 99.53% agreement (k = 0.68) for screening of the title and abstract and 88.57% agreement (k = 0.75) for screening based on the full text. Additionally, six eligible articles were identified via alternate methods. Thus, a total of 20 articles reporting 14 unique studies were included in this review [28,37,38,39,40,41,42,43,44,45,46,47,48,49,50,51,52,53,54,55]. See Appendix A for the list of excluded studies based on the full text with reasons for exclusion.

### 3.1. Characteristics of Included Studies

The included studies represented a total sample size of 329 domiciliary carers with a range of 2 to 90, although one study did not report its sample size [28,49,50]. Included studies were published between 1997 and 2023, and mostly represented England [28,38,39,40,48,49,50] and the USA [41,42,43,45], followed by Sweden [47,51], then Canada [37], Denmark [44,52], Brazil [46], Finland [53], Scotland [54], and France [55], with one study each. While all studies focused on carers of older people, in 11 studies [28,41,42,43,44,45,46,47,48,49,50,51,52,53,55] this was not defined by any specific age. One study focussed on caring for older people with alcohol-related and other dementias [37], one with alcohol-related harm [48], and one with severe drinking problems [51]. In addition, one study excluded those who were alcohol dependent [38,39,40]. Most studies focussed on formal carers, although in two studies the focus was on family caregivers [37,46]. See Table 1 for a summary of included studies.

### 3.2. Quality Appraisal

The majority of included studies were judged to be either of medium (*n* = 6) or low (*n* = 6) quality, with only two studies judged to be of high quality (see Table 2). All studies outlined the research aims, why a qualitative methodology was appropriate, and provided a clear statement of findings. Most studies included justification of the methodology chosen, details of ethical approval, and performed rigorous data analysis. However, the included studies did not always sufficiently describe how participants were sampled, provide a sample size rationale, acknowledge the role of the researcher and their relationship to participants, or discuss the implications of findings for research and practice.

Theme 1: Identification

Alcohol problems are common.

While all included studies reported domiciliary carers’ experience of alcohol use for the older adults in their care, over a third of studies reported that problems with alcohol were common among older people who required care in their home [42,43,44,47,55] across both the USA [42,43] and Europe [44,47,55]:


*“Well definitely if you count alcohol abuse, I’d say 90% of our patients have a substance abuse issue.”*
(Medicare home care nurse) [42]

These problems were exacerbated during the COVID-19 pandemic:


*“Alcohol use has become worse during COVID. We always had a lot of patients with alcoholism, but I think it became even more of a crutch, a help to deal with the pressures of COVID.”*
(Medicare home care nurse) [43]

No assessment for alcohol problems.

Over a third of included studies reported that domiciliary carers did not assess older people in their care for alcohol problems [38,42,43,48,53], and this was consistent across the USA [42,43], England [38,48], and Finland [53].


*“We do not assess SUD [Substance Use Disorder]. It is not just this agency. I have worked in six home health agencies. It was the same. I have friends who have worked in multiple agencies for decades and it is the same.”*
(Medicare home care nurse) [43]

While one English study reported that carers perceived a lack of appropriate tools to assist in the identification of older people who were drinking harmfully [48], another study [42] reported that Medicare home care nurses were aware of appropriate assessment tools but they did not make use of them as it was not a requirement of their role:


*“The patient’s substance use behaviour’s simply are not assessed…it is pretty ridiculous. The [substance] abuse issues are there…but we can’t professionally assess or treat it. I know there are approved interventions and assessment tools, but we don’t do it. Why? Well, it’s not required.”*
(Medicare home care nurse) [42]

Finally, a Finish study [53] reported that while home care professionals might assess for alcohol, this would not be reported in the care plan for fear of sigma for their client:


*“We make an assessment visit. During the visit we bring up almost everything: alcohol use, smoking, physical exercise, medication…usually we don’t write it into the plan, because it’s stigmatising. We have no right to stigmatise anybody…I myself am really cautious, I don’t want to stigmatise anybody. It’s a nasty question.”*
(home care professional) [53]

In addition, two studies [48,49] reported that domiciliary carers found it difficult to determine when normal alcohol consumption became problematic:


*“When does it [alcohol] become problematic? I don’t know how to determine this. How do you determine at what stage it is a problem?”*
(domiciliary carer) [48]

Additional overt signs of excessive alcohol use.

The majority of studies [38,44,46,47,48,49,51,53,54] reported that domiciliary carers witnessed additional overt signs of potentially harmful alcohol use within care recipients’ home environments, such as empty drink bottles [38,44,48], drinking buddies [44], falls [47,49,51], buying a lot of alcohol [49], deteriorating health [49], wandering [49], hangovers [49], fires caused by dropping cigarettes while drinking [49], a lack of hygiene [49,51], self-neglect [49,51,53], aggressive behaviour [49], failure to get out of bed [49,53], squalor [51,53], refusing entry to the home [51], incontinence [51], vomiting [51], issues with finances [53], and conflict [46,49]:


*“I mean the client really drinks a lot and is without company and doesn’t take care of the apartment at all and invoices remain unpaid, there is no food in the pantry, [the client is] unable to take care of his hygiene, and so on, I would say things are in a very bad way.”*
(home care professional) [53]

Observation, along with continuity of care were therefore deemed essential to detecting alcohol problems [47]. Three UK (England and Scotland) based studies [48,49,54] reported that by the time carers were aware of the signs of excessive alcohol use they felt that it would be too late to provide any meaningful help:


*“I call it a death wish. They don’t care if it [alcohol] is going to take their life; they have had enough.”*
(home carer) [49]

Being privy to an older person’s home environment also resulted in domiciliary carers regularly reporting being vulnerable and at risk, with some participants describing “horror stories” [45] of being subjected to abuse and feelings of discomfort or disgust [43,44,45,51,53,54,55]. Once again, these problems were all exacerbated during COVID-19 [43]:

*“I stay on my guard because people who drink can become aggressive and violent.”* (professional home caregiver) [55]

Theme 2: Management

To buy or not to buy that is the question.

Domiciliary carers in over a third of the included studies [38,45,49,51,54] raised the issue of whether it was appropriate to purchase alcohol for the older person in their care, with no clear consensus. Two studies [38,54], both UK based, reported that domiciliary carers would purchase alcohol for older people in their care:


*“You’ll take her shopping, say, on a Wednesday and buy all of it [alcohol].”*
(home carer) [38]

In contrast, two other studies based in the USA [45] and Sweden [51] reported that carers had rules (both imposed and self-imposed) against purchasing alcohol for older people in their care:


*“…try to ensure that the [care recipients] do not get booze”*
(home care worker) [51]

One English study [49] reported some carers purchasing alcohol, while others did not and reported confusion around their roles and responsibilities when it came to purchasing alcohol: *“They all said, ‘we’re not really sure’”* (home care manager) [49] with one wrong decision resulting in dismissal:


*“...I’ve just recently had to dismiss a member of staff who had just started—only lasted a week. This member of staff actually refused to buy alcohol for people and to me that is an infringement of the service users’ rights as an individual. So, my view was that worker wasn’t able to carry out their full duties.”*
(home care manager) [49]

This was despite none of the local authorities in the study having a written policy or guidance on the purchase of alcohol [49]. Domiciliary carers reported that the purchase of alcohol had a fundamental role in supporting the older person in the continuity of their lifestyle, activities of daily living, and related choices [38,49], while others reported that it prevented older people from being targeted by stores to purchase larger quantities of alcohol using the incentive of free delivery for bulk orders [54]. A lack of purchase was reported to be associated with mitigation of risks [45,49,51]. In the face of strict rules about purchasing alcohol, some carers found creative ways to allow older people in their care to have some autonomy:


*“Absolutely under no circumstances are you allowed to take them to the off-licence to buy drinks or take alcohol in for them, if they choose to get it by other means then it is up to them.”*
(home carer) [49]

One study reported taxi drivers purchasing alcohol for older people in their care [51] while another reported taking them out for a drink:


*“We actually take this gentleman out to a pub on a Monday and a Thursday, it’s like a befriending role to get him out. He’s limited to the drinks that he has.”*
(home care worker) [54]

Balancing rights and risks.

Inextricably linked to the theme “to buy or not to buy” was the theme of “balancing rights and risks”. Regardless of whether a study explicitly discussed the purchase of alcohol, the majority of studies [38,44,48,49,51,53,54,55] across a number of countries, including England [38,48,49], Scotland [54], Denmark [44], Sweden [51], Finland [53], and France, [55] reported domiciliary carers allowing older people in their care to consume alcohol as their basic human right. Studies reported individuals’ *wish* [44], *desire* [44], *choice* [44,49], and *right* [49] to consume alcohol:


*“...what we try to do is encourage all our staff to view our service users as individuals, and to have a very strong idea of service users’ rights to individual choice, and specifically where that comes into areas like alcohol, drugs, I do feel that workers with perhaps not much experience find those kind of situations quite difficult…Our view is that the service user has a right; we are not in there to make value judgements about service users and in certain ways we are there as guests of the service user in their home, and there to do obviously what they are not able to.”*
(home care manager) [49]

In addition, in half of the studies [42,43,44,47,49,54,55], carers reported allowing, and in some cases actively encouraging, alcohol use either to improve mood, negate loneliness, or both:


*“There are many vulnerable people, which influences our awareness, and then there is the alcohol, it is challenging to manage. If you feel annoyed…and one glass feels good…yeah, well, then maybe to consume is even better”*
(home care services) [44]

In one case, this resulted in the carer feeling “powerless” and in a “dilemma” over the conflict between an older person’s social life and the harmful effects of excessive alcohol use [44]. Once again, these findings were only exacerbated during the COVID-19 pandemic:


*“I think it contributed to more frustration, anger and abuse between caregivers and patients even though I think they took it [the alcohol] to relieve the anger, frustration, depression and isolation [during COVID]. That is the irony of drug dependence, isn’t it?”*
(medicare home care nurse) [43]

Concerns arose, however, regarding who would be responsible or at fault if anything happened to the older person needing care while they were drinking and was viewed, in one study, as a potential litigation risk [54]:


*“If anything happened to them you would carry the responsibility.”*
(home carer) [49]

These concerns were well founded based on previous experience and were associated with an increased risk of falls and fractures [49,51,55], accidents [49], changes to physical and mental health [41,49], and instability [49]:


*“Approximately fifteen years ago we were instructed not to buy alcohol for any client after a client drank whisky obtained by his/her home help, then fell and broke a hip.”*
(home carer) [49]

One study noted that the decision over whether to allow older people in their care to consume alcohol was often down to the individual and the context, leading to inequalities and problems in balancing demands from the family and carers with the rights of the older individual [48]. These findings illustrate the problem of balancing “rights” and “risks” within community care which emphasises client choice and autonomy [49]. One study also highlighted the consequences when choice was not considered:


*“Well, several of us were involved, including the doctor and others, and he had talked to him about disulfiram [A drug used to support the treatment of Alcohol Use Disorder by producing sensitivity to ethanol], and he [the older adult] had started to take disulfiram, but it was not his wish, so he drank while being on disulfiram…”*
(home care worker) [44]

Carers in six studies [38,45,49,51,54,55], mainly from England [38,49] and Scotland [54] but also the USA [45], France [55], and Sweden [51], reported taking action to limit risks from excessive alcohol consumption such as suggesting that their client purchase smaller amounts [38,54,55], encouraging drinking water before alcohol [38], restricting consumption once inebriated [49], making sure the client could not fall [49] or choke [51], and, as stated above, refusing to purchase alcohol [45,49,51]:


*“This male client, if I go in and he’s sitting having his can, I encourage a pint of water first…I know by the time he’s had a pint of water, he won’t want another drink.”*
(home carer) [38]

On occasion, actions taken to limit risk such as hiding alcohol [55] or adding water to alcohol [55] was taken covertly:


*“I put half a glass of wine and more water because she no longer gets up and she has a drinking problem.”*
(home caregiver) [55]

Monitor and report but do not intervene.

Despite the physical harm reduction approach described above, there was a clear and consistent finding throughout the included studies that highlighted a lack of formal intervention on behalf of domiciliary carers for alcohol problems with the older adults in their care [38,42,43,47,48,49,51,54]. Carers reported monitoring consumption and even reporting these issues to their employers, whether that be families or agencies, but felt their role did not allow them to take any further action. When alcohol problems were identified, the concern was to *“contain”* the client rather than address the alcohol problem or its underlying causes [49]. Some carers felt this resulted in them simply “enabling”, “reinforcing”, “sustaining”, or “maintaining” the problem [51]. This left carers feeling “frustrated” [43] as in many cases it meant that carers simply “watch things get worse” [42]:


*“We just tend to monitor. When we see extra bottles and things like that, we would report to the office, speak to the family and just see what they think. I mean, fair enough, we go in and look after them, but that’s where it stops. If the family is the one that you approach, it’s down to them. We haven’t really got a leg to stand on when it comes to that.”*
(home carer) [38]

Some domiciliary carers were concerned that reporting clients’ alcohol problems would damage trusting relationships [54], but reporting appeared to be more likely when domiciliary carers felt their own safety was at risk [54]. It was acknowledged that domiciliary carers were an appropriate resource for alcohol intervention with the potential for significant cost savings:


*“We don’t do the right thing as nurses because Medicare limits us. That hurts the patients and increases costs. It really is not the way to run a business and it’s not the way to care for people.”*
(Medicare home care nurses) [42]

One study reported that none of the organisations that employed the domiciliary carers had a written policy or guidance on appropriate responses to problematic drinking in older clients [49]. This resulted in problems with taking early action [48,54] and difficulties in knowing at which point a drinking problem would be prioritised, if at all [48]. Carers also reported concern as to how they might help someone who just wanted to reduce alcohol consumption rather than stopping altogether and not having sufficient time and resources (including appropriate tools) to follow up ”sensitive conversations” with individuals identified as having a problem [48]. Some carers expressed reservations about “how to have the conversation, when to raise it, how to raise it, and then what to do next” [48]. Carers described referring to external agencies due to their own lack of resources [48], but also described the need to listen, show compassion and gather a good history as key to referral to specialist services [48]. Domiciliary carers also felt they needed more time to build up trust with clients and raise alcohol issues tactfully by dropping hints rather than being direct; however, employing organisations did not expect that, as services aimed to improve efficiency by having domiciliary carers spending less, not more time with each client [54].

Maintaining the vicious circle

In many studies, domiciliary carers reported the fact that they did not assess or intervene for alcohol problems, which resulted in a vicious circle of exacerbation of alcohol use and poor health (both physical and mental), lack of compliance with other treatments, and hospital or home care admittance and readmittance [42,43,44,48,49,51,53,55].


*“With one client we are in a bad spiral of drinking. First he’s rehabilitated, sobered up and fed in hospital, and then when he returns home he takes a taxi and heads for the liquor store. After a week you find him lying on the floor and once again he’s passed to hospital. But sometimes when he’s been lying on the floor three or even four days, the home care has had to bandage his wounds.”*
(home care professional) [53]

This resulted in increased caregiver burden which often only came to an end when the home care client died:


*“She started to drop her cigarettes badly [when she was drinking]...she didn’t realise and she actually burnt to death.”*
(home carer) [49]

One study of Swedish domiciliary carers reported difficulty in interpreting the causes and effects of excessive alcohol use [47] and found it challenging to evaluate whether older people became depressed because of their increased alcohol consumption or started to use alcohol after signs of depression and anxiety had already appeared [47].

Home as a barrier to accessing support and services.

Being homebound was highlighted as a key barrier to accessing support and services for older adults with problematic alcohol use in four studies [37,42,43,54] across Canada [37], the USA [42,43], and Scotland [54]:


*“Yes, it always has been an issue. We do not treat or assess SUD [Substance Use Disorder]. So, there is no care. We can’t even help them get to outpatient substance use therapy and they are homebound!”*
(Medicare home care nurses) [43]

In one Scottish study [54], alcohol consumption was also cited as a barrier to accessing day care services for older people.

More support needed from healthcare professionals

Carers in three studies called for more support from healthcare professionals who have the knowledge, education, and resources to assist carers and older people with issues with alcohol use [37,38,53]:


*“We don’t really have much luck with GPs, to be fair. I mean, normally, if you do a referral, you will get a visit, but GPs only really tend to be interested in illnesses. [We tend to pass our concerns onto] the care manager and the district nurse, and to be referred to other people, Social Services, to see what other things can be done. I think when people come up against it, they tend to just pass it from one to the next so they haven’t got to deal with it.”*
(home carer) [38]

Theme 3: Training

Lack of alcohol education

Over a third of included studies cited a lack of carer training in relation to alcohol consumption identification and intervention [37,38,42,43,48,53]. This was consistent across Canada [37], the USA [42,43], England [48], and Europe [53]:


*“We are not required to assess for it [alcohol], our reimbursement isn’t affected, most nurses are not trained to deal with it, and the social workers, who have the best training, are so restricted by regulations that they cannot help.”*
(Medicare home care nurse) [43]

In one study, participants were motivated to attend a workshop by recognising a significant gap in their knowledge and skills in working with older adults with alcohol issues, directly impacting their confidence to engage with this group [48]. Particular areas where participants wanted to develop more confidence included working with people who had cognitive impairment or brain injury [48]. Another study reported an increasing caseload of domiciliary care clients with dementia and confusional states, which are problems that often mask or are related to alcohol use [49] and echo concerns around education, particularly in relation to alcohol-related dementia [37,38,49]:


*“I wonder what the numbers are for alcohol-related dementia in our Aboriginal community…generally, that topic isn’t well discussed... But you know, education is everything…How are we going to assess the need for families if we don’t know what we’re dealing with here?”*
(family caregiver) [37]

## 4. Discussion

The findings centred around the three main themes of identification, management, and training (see Figure 2). In terms of identification, domiciliary carers reported that problems with alcohol were common among older people who required care in their home, a situation which was only exacerbated during the COVID-19 pandemic. This is not surprising given that most of the studies included in this review were from countries, particularly the UK and the USA, where alcohol consumption among older people is rising [23,56]. However, the use of alcohol among older adults appears to vary widely across age groups and countries [57]. This variation can be partly explained both by the country-specific composition of populations and country-level contextual factors, such as the development level and alcohol prices [57]. The literature also showed a country-specific response to the COVID-19 pandemic and associated regulations, with a significant increase in the frequency of alcohol use in the USA and a significant increase in the proportion of people with problematic alcohol use in the UK [58]. Therefore, we cannot assume that these findings will be transferable to countries not included in this review.

Domiciliary carers did not assess older people in their care for alcohol problems and found it difficult to determine when normal alcohol consumption became problematic. This is surprising given there are a number of validated screening tools available. A systematic review of nine studies found that the Alcohol Use Disorders Identification Test (AUDIT) was useful for screening for hazardous alcohol use in patients older than 60 years and that the CAGE questionnaire was helpful in screening for alcohol dependence [59]. In addition, some screening tools have been developed specifically for older adults, including the Short Michigan Alcoholism Screening Test-Geriatric Version (SMAST-G) [60] and the Senior Alcohol Misuse Indicator (SAMI) [61]. Some domiciliary carers, particularly those employed by Medicare, appeared to be aware of screening tools, however screening for alcohol problems was specifically not part of their role.

Domiciliary carers were privy to the additional overt signs of excessive alcohol use within care recipients’ home environments which alerted them to their clients’ alcohol problems; however, this often put them in risky situations, which made carers feel vulnerable. A systematic review of violence towards formal and informal caregivers in the home care setting showed a high prevalence rate of violence with risk for physical and mental injuries for formal caregivers, although no information was available on violence against informal caregivers [62]. The authors suggested that strategies for prevention and intervention against violence in the home care setting must be developed [62].

The management of alcohol problems mainly focussed on whether it was appropriate to purchase alcohol for the older person in their care, with no clear consensus. Domiciliary carers often allowed older people in their care to consume alcohol as their basic human right, but concerns arose regarding who would be responsible if anything happened to the older person needing care while they were drinking. Carers reported taking action to limit the risks of excessive alcohol consumption. It is worth noting that there were examples of negative consequences either way: purchase alcohol and the carer could be seen as enabling harmful use, or do not purchase alcohol and the carer could be seen as disregarding a right to autonomy and thus not fulfilling their role. There was consensus that carers monitor and report excessive alcohol consumption but do not intervene, although it was acknowledged that domiciliary carers were an appropriate resource for alcohol intervention. Some carers expressed reservations about how to have the conversation, when to raise it, how to raise it, and then what to do next. Carers reported that as they did not assess or treat alcohol problems, this resulted in a vicious circle of the exacerbation of alcohol use and poor health. Needing care in the home was reported to be a barrier to accessing support and services, and more support from healthcare professionals was suggested.

### 4.1. Future Perspectives

Most older adults at risk of alcohol problems do not need specialised treatment or services. Many could benefit from screening and brief intervention to prevent problems with alcohol before they occur [63]. Brief alcohol interventions consist of one or more time-limited conversations to help reduce unhealthy alcohol use [64]. The Making Every Contact Count (MECC) initiative, developed out of the brief intervention literature, is broadly defined as an opportunistic approach to prevention by making use of the thousands of conversations service providers have with service users every day [65]. A recent Delphi consensus study confirmed that MECC is a person-centred approach to health behaviour change that, provided the individual possesses the relevant skills, can be delivered by anyone and anywhere [65]. Therefore, given appropriate training, domiciliary carers could be a useful resource for alcohol behaviour change. Unfortunately, carers reported a lack of alcohol education and training. Particular areas where participants wanted to develop more confidence included working with people who had cognitive impairment, brain injury, dementia, including alcohol-related dementia, and confusional states. Resourcing the additional requirements for time and training needed for MECC would need further investigation with domiciliary carers and providers.

Introducing additional requirements for home care staff, however, may be problematic, particularly in the UK given unmet care needs and staff shortages. Vacancy rates for home care are above their pre-pandemic levels, half of home care workers are on zero-hours contracts (the employer is not obliged to provide any minimum number of working hours to the employee) and despite its inclusion on the Shortage Occupation List, vacancies are high, and recruitment and retention remain challenging [66]. Pay is low [67], the chance of experiencing work-related violence is high, and delivering care in people’s homes presents additional risks to staff (e.g., lone working, substance misuse, and unsuitable environments and equipment) that would not be present in an institutional setting [68,69]. Training and support to staff to mitigate risks and providing support to domiciliary care workers who experience work-related aggression and violence are important for their wellbeing [68]. Additionally, the provision of training was one of the concurrent factors identified to aid staff retention [66].

### 4.2. Strengths and Limitations

This review involved a comprehensive search of both the published and grey literature that included studies of domiciliary carers of older adults with a variety of drinking histories, including alcohol-related harm. The qualitative literature approach was deemed most appropriate for this review which focused on domiciliary carers’ perspectives, and while qualitative research may be viewed as subjective, systematic reviews of qualitative data are a common and accepted form of review. Using methods developed over many years to robustly synthesise qualitative data, the interpretation of primary qualitative data and inferences from study authors were used to develop new thematic interpretations via thematic synthesis. While there were only 14 studies eligible for inclusion in this review, these studies were reported across 20 individual reports so that the thematic synthesis included a significant depth of data. The reporting of some of these studies led to issues with their appraised quality, and there were no translation services available, which may have affected the ability to include all eligible studies as well as the quality of included studies. As with any systematic review, there is a small possibility that some of the potentially eligible literature may have been missed. In addition, only those quotations presented within the included reports were available for extraction; thus, it is unclear whether the conclusions made in this review reflect the full datasets of the included studies.

## 5. Conclusions

Given the exposure of domiciliary carers to the signs of alcohol use, they seem well placed to make every contact count. Domiciliary carers would benefit from resources to assess alcohol consumption and the skills and tools to intervene. Further research should focus on developing clear guidelines for domiciliary carers on how to manage their clients’ alcohol consumption to harmoniously balance rights and risks, and particularly how to manage alcohol use in clients with dementia and other forms of cognitive decline. It is imperative that such guidance be endorsed by the leadership and management of domiciliary carers to help mitigate the dilemmas they face. This will be most relevant to countries where levels of alcohol consumption in older age groups are on the increase.

## Figures and Tables

**Figure 1 ijerph-21-01324-f001:**
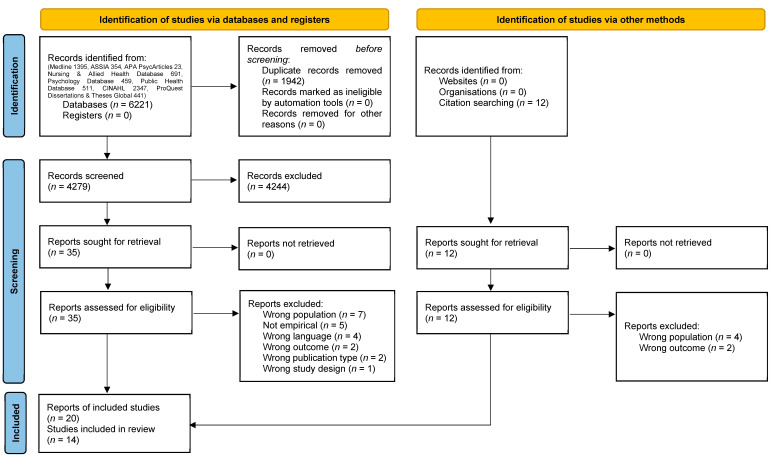
PRISMA 2020 flow diagram for new systematic reviews which included searches of databases, registers, and other sources.

**Figure 2 ijerph-21-01324-f002:**
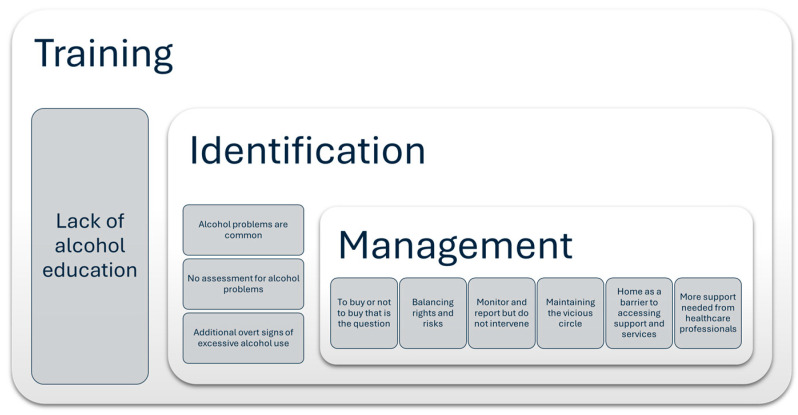
Summary of themes.

**Table 1 ijerph-21-01324-t001:** Summary of included studies.

Study/Country [References]	Aim	Sample (Relevant to Review Aims)	Methodology	Themes/Description of Findings (Identified by Authors/Relevant to Review Aims)
Alcock Canada [37]	The experiences of Indigenous female caregivers for a loved one diagnosed with Alzheimer’s disease and other dementias to examine the systemic barriers to navigating the Canadian healthcare system	Four female family domiciliary caregivers of seniors (age 55+) diagnosed with Alzheimer’s disease and other dementias, including alcohol-related dementia	Oral storytelling with qualitative thematic analysis	Support and services for seniors with regular substance use is a gap in healthcare that is discussed as a case study because healthcare providers are not addressing it due to ageism and a lack of education. This creates barriers for Indigenous and non-Indigenous seniors to access services and support
Bareham England [38,39,40]	To explore the views of older adults and primary care providers regarding health and psychosocial factors shaping drinking practices in later life, and how these practices are influenced	Two domiciliary care providers (DCPs) of adults aged 65 years or over who had consumed alcohol at some point in their lives. Individuals who were alcohol dependent or had previously engaged in treatment for alcohol misuse were excluded from the study	In-depth interviews and focus groups	DCPs were involved in care recipients’ alcohol purchase and drinking practices through their roles in supporting their daily living. DCPs reported having received no training regarding alcohol-related health risks and intervention. DCPs had a low sense of professional accountability. DCPs flagged concerns for care recipients when alcohol use was perceived to have become problematic. but had little scope for any level of intervention. DCPs had access to additional signs of potentially harmful alcohol use within care recipients’ home environments
Cabin USA [41,42]	To explore how home care nurses perceive the coverage of social determinants of health and the impact on patients, caregivers, and payers, including Medicare	Thirty-seven home care nurses of the homebound beneficiary population who were described as “mostly elderly”	Qualitative interviews using a grounded theory approach	A high frequency of substance use, abuse, and disorders. Failure to professionally assess and treat substance use issues. Lack of coverage of substance misuse and disorders exacerbates physical health, mental health, and substance use issues. The inability to provide transportation and personal care assistants limits patients’ ability to access outpatient substance treatment services. Failure to address substance use issues contributes to home care readmissions, rehospitalisations, and caregiver burden
Cabin USA [43]	To provide insight into how home care nurses perceive the impact of COVID-19 on their homebound beneficiary population	Forty-eight home care nurses of the homebound beneficiary population who were described as “mostly elderly”	Qualitative interviews using a grounded theory approach	Need for social services support increased; loneliness and depression increased among patients; physical and mental health conditions became exacerbated; substance use and abuse increased; evidence of domestic violence against patients increased; and there was a limited amount of staff and equipment to care for patients
Christiansen Denmark [44,52]	How can dealing with alcohol use in eldercare be improved to increase the quality of life of older adults and facilitate care workers?	Six home care workers, managers, and a volunteer of older adults living alone	Qualitative interviews	Conflict in alcohol-related values; alcohol as a barrier to care; integrating conflicting values to practice
Conahan USA [45]	To explore the differences in the negotiation experiences of agency and independent (subsidised participant, consumer-directed, and private pay) home care aides	Forty-nine agency (*n* = 22), independent subsidised (*n* = 9), and independent unsubsidised grey-market (*n* = 18) home care aides for older clients	Qualitative interviews	First encounters, negotiating tasks, and limiting the tasks
de Jesus Brazil [46]	To understand the relationships of relatives with the elderly person at home	Twenty family members who live together at home with elderly people	Qualitative, descriptive, and exploratory study	Conflict in relationships related to abusive use of alcohol by the elderly person: among the difficulties found in coexistence with the elderly person, the abusive use of alcohol was highlighted, which emerged as a limitation to a harmonious family relationship
Grundberg Sweden [47]	To describe home care assistants’ (HCAs) perspectives on detecting mental health problems and promoting mental health among homebound seniors with multimorbidity	Twenty-six home care assistants with any experience caring for seniors with multimorbidity	Descriptive qualitative study design using semi-structured interviews	Most HCAs stated that they were experienced in caring for clients with mental health problems such as anxiety, depression, sleep problems, and high alcohol consumption. The HCAs mentioned as causes, or risk factors, multiple chronic conditions, feelings of loneliness, and social isolation. The findings revealed that continuity of care and seniors’ own thoughts and perceptions were essential to detecting mental health problems. Observation, collaboration, and social support emerged as important means of detecting mental health problems and promoting mental health
Hafford-Letchfield England [48]	To explore and describe current issues for the community-based workforce working with older people with alcohol-related harm	Two domiciliary carers working with older people with alcohol-related harm	Workshops involving themed discussions	Difficulties mentioned around identifying alcohol problems and taking early action. Practitioners reported that they were able to spot harmful drinking behaviour if visiting people at home. This did not mean they would address alcohol consumption in their assessments. Other dilemmas arose from the need to make decisions regarding whether an older person should be refused alcohol
Herring England [28,49,50]	To examine policy and practice regarding the purchase of alcohol for older clients of home carers/to assess the current and potential role of home carers in the identification and response to problems associated with alcohol use in older people	Home care managers and home carers of elderly clients (sample size not disclosed)	Semi-structured and focus group interviews	The findings illustrated the problem of balancing “rights” and “risks” within a philosophy of community care which emphasises client choice and autonomy, and showed how policy and practice are “tailored” by local contexts
Karlsson Sweden [51]	To describe and analyse home care workers’ narratives about older people who can be characterised as heavy drinkers	Eighteen home care workers (assistant nurses or home helpers) employed in elder care and working with older people with severe drinking problems in their homes	Focus group interviews	The homes of the care recipients; descriptions of squalor. The intoxicated body; it was not just the homes of the older drinking individuals that were dirty and filthy, but also their bodies. Havoc and disruptive behaviour. Involuntarily drawn into their own world
Koivula Finland [53]	To examine how the alcohol use of elderly home care clients affects the daily work of home care professionals and how the professionals act to support the drinking client	Ten home care professionals working with community-dwelling elderly people using alcohol in their homes	Semi-structured interviews	Home care and older people’s alcohol consumption: supporting a client’s life management, professional qualifications (not trained re alcohol), and multi-professional collaboration (workers felt they needed support from other professionals
Millard Scotland [54]	To assess the needs and evaluate home care services being provided to elderly clients	Ninety staff and managers providing home, day, and residential care to elderly clients over 65 years of age	Focus groups	The trusting relationship that developed between the home care worker and the client, their perceptions regarding the client’s alcohol consumption, and the existence of barriers to involvement in day care secondary to the client’s alcohol usage
Moscato France [55]	Professional home caregivers’ perceptions of their job along with their difficulties and satisfactions in supporting older people with Alzheimer’s disease or alcohol misuse.	Seventeen professional home caregivers working in agencies or home care associations exclusively caring for older people. Those employed directly by the older person were not included in this study	Semi-structured research interviews	The main themes that emerged illustrated the nature of the associated pathologies, the perceptions and satisfactions related to the profession, their adaptive skills, the difficulties related to the life context of the older person, and the wine consumption of the latter

**Table 2 ijerph-21-01324-t002:** Quality appraisal.

Study [References]	Item 1	Item 2	Item 3	Item 4	Item 5	Item 6	Item 7	Item 8	Item 9	Item 10	Quality Rating
Alcock [37]	Y	Y	Y	Y	N	Y	N	Y	Y	N	Medium
Bareham [38,39,40]	Y	Y	Y	Y	Y	Y	Y	Y	Y	Y	High
Cabin [41,42]	Y	Y	Y	N	N	N	N	Y	Y	N	Medium
Cabin [43]	Y	Y	Y	N	N	N	N	Y	Y	N	Medium
Christiansen [44,52]	Y	Y	Y	Y	N	N	Y	Y	Y	N	Medium
Conahan [45]	Y	Y	Y	Y	Y	Y	Y	Y	Y	Y	High
de Jesus [46]	Y	Y	N	N	N	N	Y	N	Y	N	Low
Grundberg [47]	Y	Y	Y	Y	N	N	Y	Y	Y	N	Medium
Hafford-Letchfield [48]	Y	Y	N	N	N	N	N	N	Y	N	Low
Herring [28,49,50]	Y	Y	N	N	N	N	N	N	Y	Y	Low
Karlsson [51]	Y	Y	N	N	N	N	Y	N	Y	N	Low
Koivula [53]	Y	Y	N	N	N	N	N	N	Y	Y	Low
Millard [54]	Y	Y	N	N	N	N	N	N	Y	Y	Low
Moscato [55]	Y	Y	N	N	N	N	Y	Y	Y	Y	Medium

Item 1. Was there a clear statement of the aims of the research? Item 2. Is a qualitative methodology appropriate? Item 3. Was the research design appropriate to address the aims of the research? Item 4. Was the recruitment strategy appropriate to the aims of the research? Item 5. Was the data collected in a way that addressed the research issue? Item 6. Has the relationship between researcher and participants been adequately considered? Item 7. Have ethical issues been taken into consideration? Item 8. Was the data analysis sufficiently rigorous? Item 9. Is there a clear statement of findings? Item 10. How valuable is the research?

## Data Availability

The original data presented in the study are openly available in the PROSPERO international prospective register of systematic reviews (https://www.crd.york.ac.uk/prospero/display_record.php?ID=CRD42024516660 (accessed on 22 February 2024)), Rayyan (https://rayyan.ai/reviews/987359/
https://rayyan.ai/reviews/941122 (accessed on 29 September 2024)) and the FigShare/Northumbria University Research Repository (https://figshare.northumbria.ac.uk/articles/dataset/Domiciliary_carers_perspectives_on_alcohol_use_by_older_adults_in_their_care_Data_extraction_for_a_systematic_review_and_thematic_synthesis_of_qualitative_studies/26064721/1 (accessed on 20 June 2024)).

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
