# Peer review of "Domiciliary Carers’ Perspectives on Alcohol Use by Older Adults in Their Care: A Systematic Review and Thematic Synthesis of Qualitative Studies"

_ijerph, 2024, doi:10.3390/ijerph21101324_

Round 1

Reviewer 1 Report

Comments and Suggestions for Authors

The work of Haighton et al is a systematic review regarding Domiciliary Carers’ Perspectives on Alcohol Use by Older Adults in Their Care. The topic of the review is interesting and the presentation was done correctly. I would suggest the authors to integrate the following points:

- the abstract should indicate more clearly the purpose of the study;

- a more detailed description regarding the quality assessment of the selected works and the checklist used;

- a summary figure of the work could be useful for the reader;

- the paragraph "future perspectives" could improve the work

Author Response

Reviewer 1

The work of Haighton et al is a systematic review regarding Domiciliary Carers’ Perspectives on Alcohol Use by Older Adults in Their Care. The topic of the review is interesting, and the presentation was done correctly. I would suggest the authors to integrate the following points:

Thank you, we are glad that you found our manuscript interesting and correctly presented. Your suggestions have been fully integrated into the revised manuscript as described in our point-by-point response below.

- the abstract should indicate more clearly the purpose of the study;

We have added a further sentence to the abstract to more clearly indicate the purpose of the study (page 1, line 14-15).

- a more detailed description regarding the quality assessment of the selected works and the checklist used;

In order to provide more detail regarding quality assessment and the checklist used in the most concise format we have added an additional table (Table 2, page 9) with quality assessment against each checklist item as well as detail of each item in order to provide greater transparency regarding how quality ratings were developed.

- a summary figure of the work could be useful for the reader;

We have now added a summary figure visually depicting the themes and their relation to one another (Figure 2 page 17)

- the paragraph "future perspectives" could improve the work

Although we had already included a section on future perspectives it was not highlighted as such therefore, we have added a heading to signpost this section. (Page 18, line 547)

Reviewer 2 Report

Comments and Suggestions for Authors

This study covers a quite interesting topic. Namely, it examines the alcohol consumption habits of the elderly involved in domestic care.

 The choice of topic is convincing and the applied data collection methodology is appropriate. The conclusions are supported by the results.

 The authors use a wide range of technical literature connected to the topic. However, It would be worth emphasizing more the fact that the characteristics of the social relationships of elderly people as well as their occasional inactive lifestyle shows a close relevance to alcohol consumption. Thus backing up the current nature of the topic. 

 In summary, the authors have covered a very important topic. The outstanding significance of the article lies in pointing out the following: The (mainly professional) domestic care system could function as a primary indicative service in recognising and treating elderly people's alcohol problems and in this way, in improving their life quality.

 I have made a plagiarism check with Turnitin. Without 'References' the text shows a 26% equality which is mainly an unavoidable consequence of presenting the results of the qualitative data collection. That is, citations from  the interviews.

Author Response

Reviewer 2

This study covers a quite interesting topic. Namely, it examines the alcohol consumption habits of the elderly involved in domestic care.

Thank you we are glad that you found our manuscript interesting.

The choice of topic is convincing and the applied data collection methodology is appropriate. The conclusions are supported by the results.

Once again thank you for your support.

The authors use a wide range of technical literature connected to the topic. However, It would be worth emphasizing more the fact that the characteristics of the social relationships of elderly people as well as their occasional inactive lifestyle shows a close relevance to alcohol consumption. Thus backing up the current nature of the topic.

Thank you for this suggestion. We have now included an extra paragraph in the introduction describing the relationship between alcohol consumption and social/lifestyle factors as follows:

“Research has also shown that older adults use alcohol for a number of different reasons [10]. For example, stressful life events, such as bereavement or retirement, may trigger late-onset drinking in some, [11, 12]. Alcohol use in older adults has also been associated with self-medication for both physical and mental health problems [13] as well as insomnia and has also been linked to boredom, loneliness, isolation and homelessness [11]. However, the direction of causality in the relationship between alcohol use and many of these factors is often in doubt [10] and clearly, the relationship between alcohol use and health is complex.” (Page 2 lines 58-65)

In summary, the authors have covered a very important topic. The outstanding significance of the article lies in pointing out the following: The (mainly professional) domestic care system could function as a primary indicative service in recognising and treating elderly people's alcohol problems and in this way, in improving their life quality.

We are glad you recognised the importance and significance of our work and agree with our conclusions.

I have made a plagiarism check with Turnitin. Without 'References' the text shows a 26% equality which is mainly an unavoidable consequence of presenting the results of the qualitative data collection. That is, citations from the interviews.

As you have rightly identified in order to support our findings from the systematic review, we have included qualitative data from the included studies however all direct quotations have been appropriately referenced.

Reviewer 3 Report

Comments and Suggestions for Authors

This article conducts a systematic review of qualitative studies of home caregivers in relation to alcohol use in older adults. Comments on the article are:

The abstract needs several changes in accordance with comments on the article.

There is a bias in the introduction regarding the comments and references indicated about the social and health benefits of low alcohol consumption. Other studies indicate the opposite. For example, see the books and reports of the World Health Organization.

The use of qualitative data for the systematic review is a problem because they are few studies, do not include quantitative data, and are more likely to introduce the subjectivity of the authors of the paper to the reviewers. As the authors comment, of the 14 articles entered in the review, only two are of high quality.

The presentation of the results is adequate for a qualitative review.

The discussion needs to be rewritten in several parts because it does not reflect the results obtained and includes several comments unrelated to the study and the results (e.g., regarding prevention and/or treatment of persons at risk for alcohol problems). The discussion should be grounded in the results and in the scientific literature on the subject of alcohol.

The paper does not include the limitations of the study. It is pertinent to include as one of the limitations the use of qualitative studies.

Author Response

Reviewer 3

This article conducts a systematic review of qualitative studies of home caregivers in relation to alcohol use in older adults. Comments on the article are:

We have responded to your comments below.

The abstract needs several changes in accordance with comments on the article.

The abstract has been amended to reflect reviewers’ comments.

There is a bias in the introduction regarding the comments and references indicated about the social and health benefits of low alcohol consumption. Other studies indicate the opposite. For example, see the books and reports of the World Health Organization.

Thank you for this comment. We have now included more detail of the negative effects of alcohol as follows:

“Alcohol is a risk factor for cancers, coronary heart disease, high blood pressure, liver cirrhosis, pancreatitis and stroke [3]. As people age, physiological changes mean that that older adults are more sensitive to the effects of alcohol, and experience problems at lower levels of consumption [4]. Alcohol and medication interactions are common for this age group [5, 6] with the average person aged 50 or above taking at least four prescribed medications a day [7] and with alcohol a major contraindication for many of these drugs. Combined alcohol and medication use is estimated to affect up to 19% of older Americans [8] while drinking alcohol for medicinal purposes is also prevalent [6]. Excessive alcohol consumption has also been associated with impairments in the instrumental activities of daily living [9] and can contribute to the onset of dementia and other age-related cognitive deficits [10], Parkinson’s disease and a range of psychological problems including depression and anxiety [11]. Alcohol use is implicated in one-third of all suicides in the older population [12].” (Page 1-2, Lines 39-52)

The use of qualitative data for the systematic review is a problem because they are few studies, do not include quantitative data, and are more likely to introduce the subjectivity of the authors of the paper to the reviewers. As the authors comment, of the 14 articles entered in the review, only two are of high quality.

Thank you for this comment. We have amended the limitations section to reflect some of these issues. The limitations section now reads as follows:

“This review involved a comprehensive search of both published and grey literature that included studies of domiciliary carers of older adults with a variety of drinking histories, including alcohol related harm. Qualitative literature was deemed most appropriate for this review which focused on domiciliary carers perspectives and while qualitative research may be viewed as subjective, systematic reviews of qualitative data are a common and accepted form of review. Using methods developed over many years to robustly synthesise qualitative data, interpretation of primary qualitative data and inferences from study authors were used to develop new thematic interpretations via thematic synthesis. While there were only 14 studies eligible for inclusion in this review these studies were reported across 20 individual reports so that the thematic synthesis included a significant depth of data. The reporting of some of these studies led to issues with their appraised quality and there were no translation services available, which may have affected the ability to include all eligible studies as well as the quality of included studies. As with any systematic review there is a small possibility that some potentially eligible literature may have been missed. In addition, only those quotations presented within the included reports were available for extraction thus it is unclear whether the conclusions made in this review reflect the full datasets of included studies.” (page 19). More detail of the quality appraisal process has also been included in the manuscript (see Table 2).

The presentation of the results is adequate for a qualitative review.

Thank you for identifying that the presentation of the results is appropriate for a systematic review of qualitative studies.

The discussion needs to be rewritten in several parts because it does not reflect the results obtained and includes several comments unrelated to the study and the results (e.g., regarding prevention and/or treatment of persons at risk for alcohol problems). The discussion should be grounded in the results and in the scientific literature on the subject of alcohol.

Thank you for this comment. We have kept the discussion section grounded in the results and following your, and another reviewer’s, comment have moved the section regarding prevention into a section titled “future perspectives” in order to denote the suggestion regarding prevention are authors suggestions for further development (Page 18, Line 547)

The paper does not include the limitations of the study. It is pertinent to include as one of the limitations the use of qualitative studies.

Thank you for this suggestion, however the manuscript already included a strengths and limitations section and included quality of included studies as a limitation. However following your earlier comment regarding qualitative systematic reviews were have expanded the limitation section. The limitations section now reads as follows:

“This review involved a comprehensive search of both published and grey literature that included studies of domiciliary carers of older adults with a variety of drinking histories, including alcohol related harm. Qualitative literature was deemed most appropriate for this review which focused on domiciliary carers perspectives and while qualitative research may be viewed as subjective, systematic reviews of qualitative data are a common and accepted form of review. Using methods developed over many years to robustly synthesise qualitative data, interpretation of primary qualitative data and inferences from study authors were used to develop new thematic interpretations via thematic synthesis. While there were only 14 studies eligible for inclusion in this review these studies were reported across 20 individual reports so that the thematic synthesis included a significant depth of data. The reporting of some of these studies led to issues with their appraised quality and there were no translation services available, which may have affected the ability to include all eligible studies as well as the quality of included studies. As with any systematic review there is a small possibility that some potentially eligible literature may have been missed. In addition, only those quotations presented within the included reports were available for extraction thus it is unclear whether the conclusions made in this review reflect the full datasets of included studies.” (page 19).

Round 2

Reviewer 1 Report

Comments and Suggestions for Authors

This is a good paper. The authors have addressed the reviewer comments. I recommend publication

Comments on the Quality of English Language

No issue detected

Reviewer 3 Report

Comments and Suggestions for Authors

Accept in present form.